# A second genetic screen for *gurken* mRNA mislocalisation uncovers novel phenotypes of piRNA pathway mutants in *Drosophila*

Sophie J. Liddell[1,‡], Rahnya Taghi[2,§], Jessie-Siling Li[2,§], Sejal Sathe[2], Shashank Chary[2], Azusa Hayashi[2], S. Mark Wainwright[1,3], Sheena Pinchin[1], David Ish-Horowicz[1,4,*] and Rippei Hayashi[1,2,‡,¶]

## ABSTRACT

Transposon silencing is essential for germline development. In *Drosophila* oogenesis, DNA damage caused by transposon activation affects microtubule-dependent mRNA localisation in the oocyte and impairs embryonic axes formation. Our previous EMS mutagenesis screen for *gurken* mRNA mislocalisation on chromosome 3L of *Drosophila melanogaster* identified several piRNA pathway mutants (Hayashi et al., 2014). Here, we report the screen for chromosome 3R. We identified ten mutation groups disrupting *gurken* mRNA localisation and other mutations affecting different aspects of oogenesis. We found that mutations in *karyopherin-β3* affect localisation and translation of *gurken* mRNA in a transposon silencing-independent manner. Characterisation of the new mutation in *vreteno* revealed that the piRNA pathway is essential for the basal stalk development, the process of holding the ovariole and encapsulating the first egg chambers. Females transheterozygous for *vreteno* and *armitage* mutations both showed abnormal basal stalks, defective egg chamber formation and loss of germline cells. We also found that the mutation in the Zinc Finger motif of Spindle-E shows a hypomorphic transposon activation phenotype, consistent with the previous study (Ott et al., 2014). Further characterisation showed that the Zinc Finger is required for robust ping-pong piRNA biogenesis and the nuage localisation of AGO3, but not of Aubergine, suggesting that it is involved in a specific step of ping-pong biogenesis.

KEY WORDS: piRNA pathway, Transposon silencing, EMS mutagenesis screen, *Drosophila* oogenesis, *gurken* mRNA localisation

[1]Developmental Genetics Laboratory, London Research Institute, Cancer Research UK, 44 Lincoln's Inn Fields, London WC2A 3LY, UK. [2]The Shine-Dalgarno Centre for RNA Innovation, The John Curtin School of Medical Research, Australian National University, Canberra, ACT 2601, Australia. [3]Department of Physiology, Anatomy and Genetics, Sherrington Building, Parks Road, University of Oxford, Oxford OX1 3PT, UK. [4]MRC Laboratory for Molecular Cell Biology, University College London, Gower Street, London WC1E 6BT, UK.
*Deceased
‡These authors contributed equally to this work
§These authors contributed equally to this work

¶Author for correspondence (rippei.hayashi@anu.edu.au)

 R.H., 0000-0002-5848-9019

## INTRODUCTION

Germline development requires coordination of cell growth, differentiation and meiosis. For example, in *Drosophila* oogenesis, meiosis arrests at prophase I after meiotic DNA recombination. Ensuing double-stranded DNA breaks are repaired before the oocyte starts to grow and undergo differentiation. Failure in fixing the DNA damage or excessive DNA damage activates the checkpoint kinase and perturbs processes of oocyte development, such as microtubule-dependent mRNA transport (Ghabrial and Schüpbach, 1999).

mRNA transport occurs in two distinct developmental stages during *Drosophila* oogenesis (Lasko, 2012). First, certain RNAs, such as *oskar* mRNA, are selectively transported from undifferentiated nurse cells to the presumptive oocyte soon after the germline stem cell divides four times to form a cyst that consists of 16 interconnected germline cells. Defective mRNA transport at this early stage results in loss of the oocyte, the phenotype seen, for example, in *egalitarian* (16 equal nurse cells) (Mach and Lehmann, 1997). mRNA transport from nurse cells to the oocyte continues to occur throughout oogenesis. Distinct and partially common mechanisms further sort selected mRNAs within the oocyte (Bullock and Ish-Horowicz, 2001). These asymmetrically localised mRNAs during mid to late oogenesis play crucial roles in determining the antero-posterior and dorso-ventral axes of the oocyte and the embryo as well as in forming the germ plasm that is later required for specifying the primordial germ cells in the embryos (MacDougall et al., 2003; Gáspár et al., 2017).

Early works by Trudi Schüpbach and her colleagues identified maternal mutations that disrupt *Drosophila* eggshell polarisation (Schüpbach and Wieschaus, 1991). Those that show a lack of dorsal appendages were collectively called spindle mutations, and all commonly affect *gurken* mRNA localisation and translation (González-Reyes et al., 1997). *gurken* encodes TGF-α like ligands that signal the somatic follicle epithelial cells at the dorsal side of the egg chamber to instruct appendage formation (Schüpbach, 1987). Genetic and biochemical characterisation of the spindle mutations attributed *gurken* mRNA mis-localisation and abnormal microtubule organisation to the DNA damage response, typically caused by transposon activation and loss of the Piwi-interacting RNA (piRNA) pathway (Chen et al., 2007; Klattenhoff et al., 2007).

piRNA is a class of small RNAs that is deeply conserved in animals (Czech et al., 2018). piRNAs are loaded onto the PIWI-clade Argonaute proteins to form piRNA-induced silencing complexes. Of the three PIWI proteins expressed in *Drosophila* ovaries, Aubergine and AGO3 are exclusively expressed in the germline cells while Piwi is expressed in both germline and somatic cells (Brennecke et al., 2007). Aubergine and Piwi carry piRNAs that are derived from the antisense strand of transposons and elicit post-transcriptional and transcriptional silencing, respectively (Sato and Siomi, 2020). Loss of

genes for the germline piRNA pathway affects *gurken* mRNA localisation and dorsal appendage formation, while loss of genes for the somatic piRNA pathway affects egg chamber formation and ovary morphology via a poorly understood mechanism (Hayashi et al., 2014; Zamparini et al., 2011).

We previously reported an EMS mutagenesis screen for chromosome 3L based on scoring the mis-localisation of fluorescently labelled *gurken* mRNA in germline clones made by FRT-driven mitotic recombination and the dominant female sterile allele, $ovo^{D1}$ (Hayashi et al., 2014). This screen recovered mutations in the piRNA pathway genes, *armitage* and *maelstrom*, as well as other mutations that directly affect microtubule-dependent cargo transport. Since we scored the phenotype in mitotic germline clones in otherwise mosaic females, the screen allowed us to recover lethal mutations as well as mutations affecting gross ovary morphology when they become homozygous. For example, while classical alleles of *armitage* only affect germline expression of *armitage*, the *armitage* alleles identified in the screen showed defective ovarian development (Cook et al., 2004). Here, we report an analogous screen targeted for chromosome 3R. We identified mutations in known piRNA pathway genes, *spindle-E*, *vreteno*, *qin/kumo*, *minotaur* and *kotsubu* as well as a mutation in *hyrax*, a novel gene for transposon silencing. Characterisation of *vreteno* and *armitage* mutations revealed a process of pupal ovary development that is commonly affected by mutations in the somatic piRNA pathway components. Furthermore, we serendipitously recovered a second mutation in the Zinc Finger motif of Spindle-E and confirmed the hypomorphic transposon overexpression phenotype (Ott et al., 2014). We showed that the Zinc Finger motif is required for robust ping-pong piRNA biogenesis. Thus, this mutagenesis screen revealed novel phenotypes of piRNA pathway mutants that were not identified by previous cell type-specific RNAi screens.

## RESULTS AND DISCUSSION
### The screen recovered mutations affecting *gurken* mRNA localisation and other aspects of oogenesis
We conducted the genetic screen between 2010 and 2013 in the Ish-Horowicz laboratory as part of S.J.L.'s PhD thesis (Fig. 1). Selected

mutations were kept and further characterised by R.T. et al. Briefly, we individually mutagenised about 7000 3R chromosomes and successfully scored the phenotype of ovaries for about 5000 chromosomes. We found that 116 chromosomes repeated a phenotype twice, and they were subsequently tested for complementation. Seventy-two chromosomes that showed either *gurken* mRNA mis-localisation in the mid-stage oocytes or the nurse cell dumpless phenotype were grouped together because the latter often showed the former phenotype to a varying degree. We recovered ten complementation groups that primarily showed a *gurken* mRNA mis-localisation phenotype and ten groups that showed other miscellaneous phenotypes (Tables 1 and 2). We established complementation groups by making transheterozygous crosses between the mutated chromosomes as well as with known mutations on chromosome 3R (summarised in Tables 1 and 2). Mutations were mapped to roughly one cytogenetic unit by meiotic recombination mapping and allele-discriminating PCR utilising single nucleotide polymorphisms across the chromosome. Causative mutations were finally identified by sequencing the genomic DNA extracted from homozygous 1st instar larvae (Table 3, see Materials and Methods).

### *hyrax* is required for transposon silencing
Of the ten complementation groups that showed *gurken* mRNA mis-localisation phenotype, *hyrax* and *karyopherin-β3* were not previously characterised. We only recovered one allele for *hyrax*; strong alleles of *hyrax*, $hyrax^1$ and $hyrax^2$, did not produce scorable clones, likely due to their cell lethality (Mosimann et al., 2006). Therefore, we made a second chromosome of the same mutation (Gly314>Asp) by CRISPR-Cas9 editing (see Materials and Methods). Both the original *M23* allele and the CRISPR allele *2-m5-5* were pupal lethal over $hyrax^1$. We examined expression of *Burdock* retrotransposon, which is normally suppressed but is upregulated in piRNA pathway mutants (Senti et al., 2015). Fluorescent RNA *in situ* hybridisation showed overexpression of *Burdock* mRNAs in both $hyrax^{M23}$ and $hyrax^{2-m5-5}$ mutant clones, suggesting that the *gurken* mRNA mis-localisation was caused by transposon activation (Fig. 2A). *hyrax* encodes the *Drosophila*

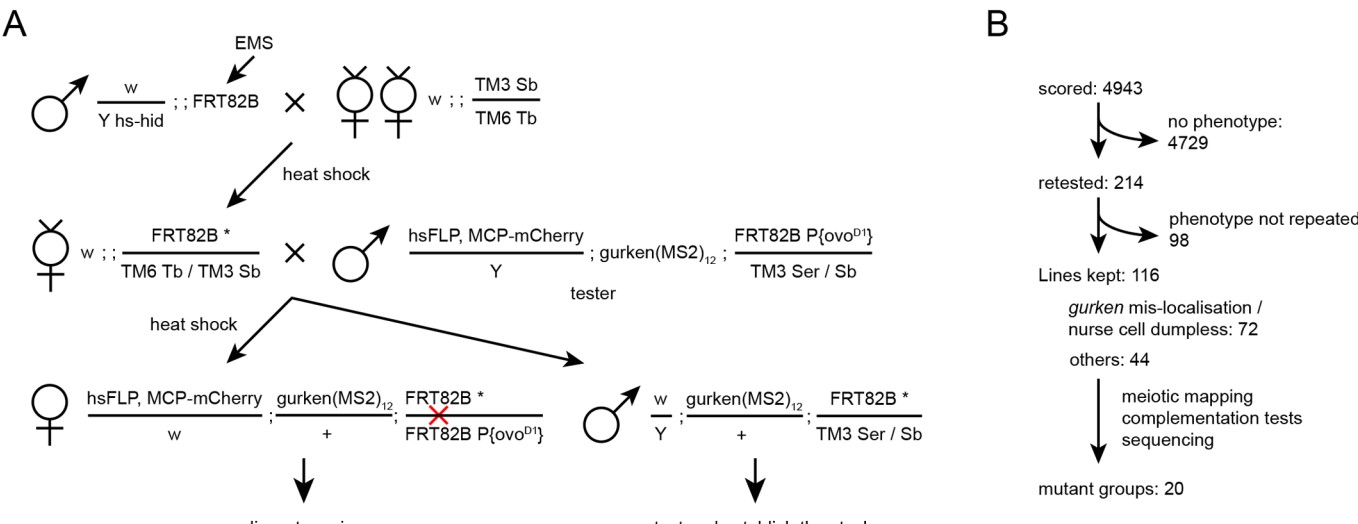

**Fig. 1. Workflow of the chromosome 3R *gurken* mRNA mislocalisation screen.** (A) Asterisk indicates the mutagenised *FRT82B* chromosome. *hs-hid* males are selectively eliminated by heat shock during larval stages. The red cross indicates a FLP-induced mitotic recombination event between the two FRT sequences. All developing egg chambers are homozygous for induced mutations on 3R, because they lack the dominant female-sterile $ovo^{D1}$ gene. (B) Shown are the number of mutated chromosomes examined at each step of the screen.

**Table 1. List of mutations affecting *gurken* mRNA localisation in late-stage oocytes**

| Gene | Alleles | Other phenotypes | Transheterozygous/hemizygous phenotypes |
|---|---|---|---|
| spindle-E | P8*, P53*, P9*, P55*, P64*, M525* | TE desilencing[#,1], egg ventralisation | female sterile |
| vreteno | P38* | TE desilencing[#,2], egg ventralisation | female sterile, confirmed by vreteno[D1] |
| hyrax | M23* | TE desilencing[#], egg ventralisation | pupal lethal, confirmed by hyrax[1] |
| qin/kumo | P69 | TE desilencing[3], egg ventralisation | |
| minotaur/CG5508 | R33, M6 | TE desilencing[4], egg ventralisation | female sterile |
| kotsubu/CG9925 | P11 | weak egg ventralisation | female semi-sterile, confirmed by Df(3R)/BSC616 |
| spindle-A | 141M | DNA damage[5], egg ventralisation | female sterile, confirmed by spindle-A[1] |
| spindle-F | S53 | DNA damage[6], egg ventralisation | female sterile, confirmed by spindle-F[1] |
| squid | R6 | egg dorsalisation[7] | lethal, confirmed by squid[1] |
| karyopherin-b3 | P14*, P56*, M15 | reduced Gurken translation[#], egg ventralisation | female sterile |

Mutations marked by hash tags were further investigated in this study. Alleles marked by asterisks were kept in the Hayashi lab. These phenotypes were previously reported: [1](Klenov et al., 2007), [2](Handler et al., 2011), [3](Anand and Kai, 2012), [4](Vagin et al., 2013), [5](González-Reyes et al., 1997), [6](Abdu et al., 2006), [7](Kelley, 1993).

homologue of Cdc73, the core subunit of the RNA polymerase II associated factor 1 (PAF1) complex. The PAF1 complex and Cdc73 are highly conserved from yeast to humans. The C-terminal half of Cdc73 forms a Ras-like domain that is responsible for forming a complex with the other core subunits of the PAF1 complex (Amrich et al., 2012). Gly314 is located in the N-terminal half. The N-terminal half is known to interact with the transcription factor β-catenin (Mosimann et al., 2006), but it is absent in yeast and plants. Cdc73 is also known to interact with other transcription factors such as Cubitus interruptus (Ci)/Gli, NOTCH and Ets1 (Mosimann et al., 2009; Melnick et al., 2023). The G314D mutation likely affects TF-specific transcription involved in transposon silencing; however, the exact nature of its effects remains unclear.

### *karyopherin-β3* is required for *gurken* mRNA localisation and translation in a DNA damage-independent manner

In contrast to the *hyrax* mutations, *karyopherin-β3* mutations did not elevate the expression of *Burdock* (Fig. 2A). The oocyte genomic DNA forms a dense spherical structure called karyosome. DNA damage caused by transposon activation disturbs the karyosome formation. Neither mutation of *karyopherin-β3* affected the oocyte karyosome, indicating the absence of DNA damage (Fig. 2A′). Fluorescent *in situ* hybridisation against endogenous *gurken* mRNA in *karyopherin-β3* mutant clones confirmed the mis-localisation phenotype. *gurken* mRNA is tightly localised to the plasma membrane of the dorsal anterior corner of the mid-stage wild-type oocyte, whereas its localisation is more diffused around the nuclei of the mutant oocytes (Fig. 3C). The phenotype was reminiscent of the *orb^mel^* mutation where *gurken* mRNA mis-localisation is associated with a lack of translation in the absence of

DNA damage (Chang et al., 2001). Indeed, immuno-staining of Gurken proteins showed reduced expression and more diffuse localisation of Gurken in the *karyopherin-β3* mutant oocytes compared to a strong and more focused localisation of Gurken to the dorsal anterior corner of wild-type oocytes (Fig. 3D). Furthermore, the oocyte nuclei are frequently mispositioned to the posterior end in *karyopherin-β3* mutants, suggesting that Gurken translation is also affected in the early stages of oogenesis (Fig. 3C). Karyopherin-β3 is the *Drosophila* paralogue of yeast Kap121p, the adapter protein involved in nuclear import of a range of cargo proteins (Kobayashi and Matsuura, 2013). Crystal structures with several known nuclear localisation signals (NLSs) showed a pocket within the HEAT repeats that captures the NLSs (Kobayashi and Matsuura, 2013) (Fig. 3A). An Alphafold-predicted structure of Karyopherin-β3 revealed that the two mutations from the screen are found in the NLS-binding pocket, with one mutation altering the amino acid that directly faces the NLS peptide (Fig. 3B). Which cargo molecules are imported by Karyopherin-β3 to the nuclei of the oocyte/nurse cells, and how they are involved in *gurken* mRNA localisation and translation remain to be elucidated.

### Mutations affecting miscellaneous aspects of oogenesis

The screen recovered many mutations showing phenotypes that are not related to *gurken* mRNA localisation. We found that mutations in *arp1*, which encodes the core component of the Dynein cofactor Dynactin (Urnavicius et al., 2015), affect oocyte specification as revealed by the lack of accumulation of the oocyte marker Orb (Fig. 4A). The phenotype is consistent with previous studies that demonstrated the essential role of Dynein-dependent transport in oocyte specification (Hayashi et al., 2014; Navarro et al., 2004).

**Table 2. List of mutations affecting other aspects of oogenesis**

| Gene | Alleles | Other phenotypes | Transheterozygous/hemizygous phenotypes |
|---|---|---|---|
| arp1 | R94*, R110* | oocyte mis-specification[#] | embryonic lethal |
| ubpy/Usp8 | M4* | abnormal ring canal formation[#] | pre-pupal lethal, confirmed by ubpy[KO] |
| Delta | P33, P35 | fused egg chambers[1] | embryonic lethal |
| bag of marbles | R54 | mis-differentiated germline cells[2] | female sterile, confirmed by Df(3R)BSC848 |
| EMC1/CG2943 | P1*, M55*, S25 | nurse cell dumpless[#] | pre-pupal lethal |
| sec8 | R312*, P68*, R27 | nurse cell dumpless | pre-pupal lethal |
| cheerio | P2, R8, R12, M113, M919 | nurse cell dumpless[3] | female sterile |
| Tsc-1 | R1, R18, R586 | overgrowth, mis-positioned oocytes | pre-pupal lethal, confirmed by Tsc-1[Q87X] |
| modulo | P39, M19 | disorganised nurse cells | pre-pupal lethal |
| ferrochelatase | R3*, P10*, P16* | accumulation of protoporphyrin[#] | pre-pupal or pupal lethal |

Mutations marked by hash tags were further investigated in this study. Alleles marked by asterisks were kept in the Hayashi lab. These phenotypes were previously reported: [1](López-Schier and St Johnston, 2001), [2](McKearin and Spradling, 1990), [3](Robinson et al., 1997).

**Table 3. Nucleotide and amino acid changes identified in the screen**

| Gene | Protein ID | Nucleotides/amino acid changes |
|------|-----------|-------------------------------|
| spindle-E | FBpp0082637 | P8: Gln72 (CAG)>STOP (TAG) |
| | | P9: Cys1012 (TGC)>Tyr (TAC) |
| | | P53: Gln861 (CAG)>Stop (TAG) |
| | | P55: Gln785 (CAG)>Stop (TAG) |
| | | P64: Cys1400 (TGC)>Tyr (TAC) |
| | | M515: AG>AA, splice acceptor of intron 8, exon 9 will be skipped, causing a frameshift in exon 10 |
| vreteno | FBpp0083716 | P38: Gln34 (CAG)>Stop (TAG) |
| hyrax | FBpp0081448 | M23: Gly314 (GGC)>Asp (GAC) |
| qin/kumo | FBpp0302955 | P69: Gln1126 (CAG)>Stop (TAG) |
| minotaur/CG5508 | FBpp0307019 | R33: Trp112 (TGG)>Stop (TAG) |
| | | M6: Gln158 (CAG)>Stop (TAG) |
| kotsubu/CG9925 | FBpp0082339 | P11: Gln570 (CAG)>Stop (TAG) |
| karyopherin-b3 | FBpp0307568 | P14: Ser428 (TCC)>Phe (TTC) |
| | | P56: Ala465 (GCC)>Val (GTC) |
| arp1 | FBpp0082121 | R94: Gly302 (GGC)>Asp (GAC) |
| | | R110: Gln229 (CAG)>Stop (TAG) |
| ubpy/Usp8 | FBpp0083419 | M4: Asn727 (AAT)>Lys (AAA) |
| EMC1/CG2943 | FBpp0308770 | M55: Gln748 (CAA)>Stop (TAA) |
| | | P1: 7 nucleotides deletion: AGCATTGTCCACTCCGAAAACTGGCTT>AGCATTgcaagcacTGGCTT, causing a frameshift from Val707 |
| sec8 | FBpp0078326 | R312: Arg244 (CGA)>Stop (TGA) |
| | | P68: Leu692 (TTG)>Stop (TAG) |
| cheerio | FBpp0428261 | M113: Trp148 (CAG)>Stop (TAG) |
| | | M919: AG >TG, splice acceptor of intron 14; exon 15 will be skipped in Cher-PO, causing a frameshift in exon 16 |
| modulo | FBpp0085233 | M19: Gln69 (CAA)>Stop (TAA) |
| | | P39: large deletion before telomere including the gene |
| ferrochelatase | FBpp0085208 | P10: Tyr330 (ACC)>Ile (ATC) |
| | | P16: Gly38 (GGT)>Asp (GAT) |
| | | R3: GTGAGT to GTGAGA, six nucleotides downstream of exon 3, which likely interferes with splicing |

Numbers correspond to the position of amino acid residues in the coding sequences found in the FlyBase protein ID. Underlining indicates nucleotide changes.

We identified a single mutation in *ubpy/Usp8* causing membrane rupture of the germline cells in early to mid-oogenesis (Fig. 4B). *Ubpy* encodes a deubiquitinase that is involved in ESCRT-dependent membrane trafficking. The *M4* mutation as well as the knock-out deletion mutation of *ubpy* (Mukai et al., 2010) caused a membrane loss between the nurse cells as shown by faint staining of phalloidin, which resulted in aggregation of the nurse cell nuclei and an abnormal ring canal structure (Fig. 4B).

Three mutations identified in the screen caused an accumulation of red fluorescent substances in germline cells, thereby masking *gurken* mRNA that was marked by mCherry (Fig. 1). Sequencing identified mutations in *ferrochelatase*, which encodes the terminal enzyme of heme biosynthesis. Ferrochelatase inserts ferrous iron into protoporphyrin IX to generate protoheme IX (heme) (Wu et al., 2001). The mutations in *ferrochelatase* caused pre-pupal/pupal lethality likely due to depletion of heme. The accumulation of the heme precursor protoporhyrin IX is the probable source of red fluorescent substances observed in our screen and in larvae transheterozygous for the *ferrochelatase* mutations (Fig. 4C).

The screen recovered three mutations in *CG2943*, which encodes the *Drosophila* homologue of ER membrane complex subunit 1 (EMC1), showing the nurse cell dumpless phenotype in late-stage egg chambers (Fig. 4D). Nurse cells start expelling their cytoplasmic materials to the oocyte through ring canals as they undergo apoptosis in stage 10b egg chambers, a process called nurse cell dumping. A network of actin filament cables forms during this process to trap nurse cell nuclei to prevent blockage at ring canals (Guild et al., 1997). We found that *EMC1/CG2943* mutants fail to retain the nurse cell nuclei during the dumping events (Fig. 4D). The EMC is a transmembrane domain insertase, which inserts multi-pass integral membrane proteins as well as tail-anchored proteins to the ER membranes (Guna et al., 2018; Shurtleff et al.,

2018). Interestingly, our previous screen on 3L recovered nurse cell dumpless mutations in the *Drosophila* homologue of Chd5/WRB/GET1, the ER membrane receptor for the tail-anchored proteins (Hayashi et al., 2014). Although two mutations independently provided the genetic link, the molecular link between ER membrane protein insertion and nurse cell dumping remains mysterious.

**Ovarian somatic piRNA pathway is required for basal stalk formation**

Mutations affecting *gurken* mRNA localisation provided a unique opportunity to investigate the function of transposon silencing in oogenesis. We previously showed that females transheterozygous for strong amorphic alleles of *armitage* do not lay eggs due to germline loss and abnormal egg follicle formation (Hayashi et al., 2014). The *vreteno* mutation *P38* from the screen also caused the 'no eggs' phenotype when it was crossed with the other null allele *vreteno^{delta1}* (Handler et al., 2011). Both *armitage* and *vreteno* are required for producing primary piRNAs in the germline and the somatic cells (Handler et al., 2011; Saito et al., 2010). Abnormal ovarian follicle phenotype was also reported for other mutations of *vreteno* and of *eggless*, which encodes the Histone H3 Lysine 9 methyltransferase involved in Piwi-mediated transposon silencing (Zamparini et al., 2011; Clough et al., 2007; Sienski et al., 2015). These observations indicate a developmental role of the ovarian somatic piRNA pathway for protecting the ovaries from transposon activation.

We dissected pupal ovaries to investigate earlier defects of oogenesis when the ovariole structure begins to form. We determined pupal stages based on morphological features such as Malpighian tubules, the yellow body (transient pupal midgut), orbital vibrissae, and abdominal bristles (Bainbridge and Bownes, 1981) (Fig. S1). In pupal ovary development, the first egg chamber forms at around 30 to 60 h after puparium formation (APF)

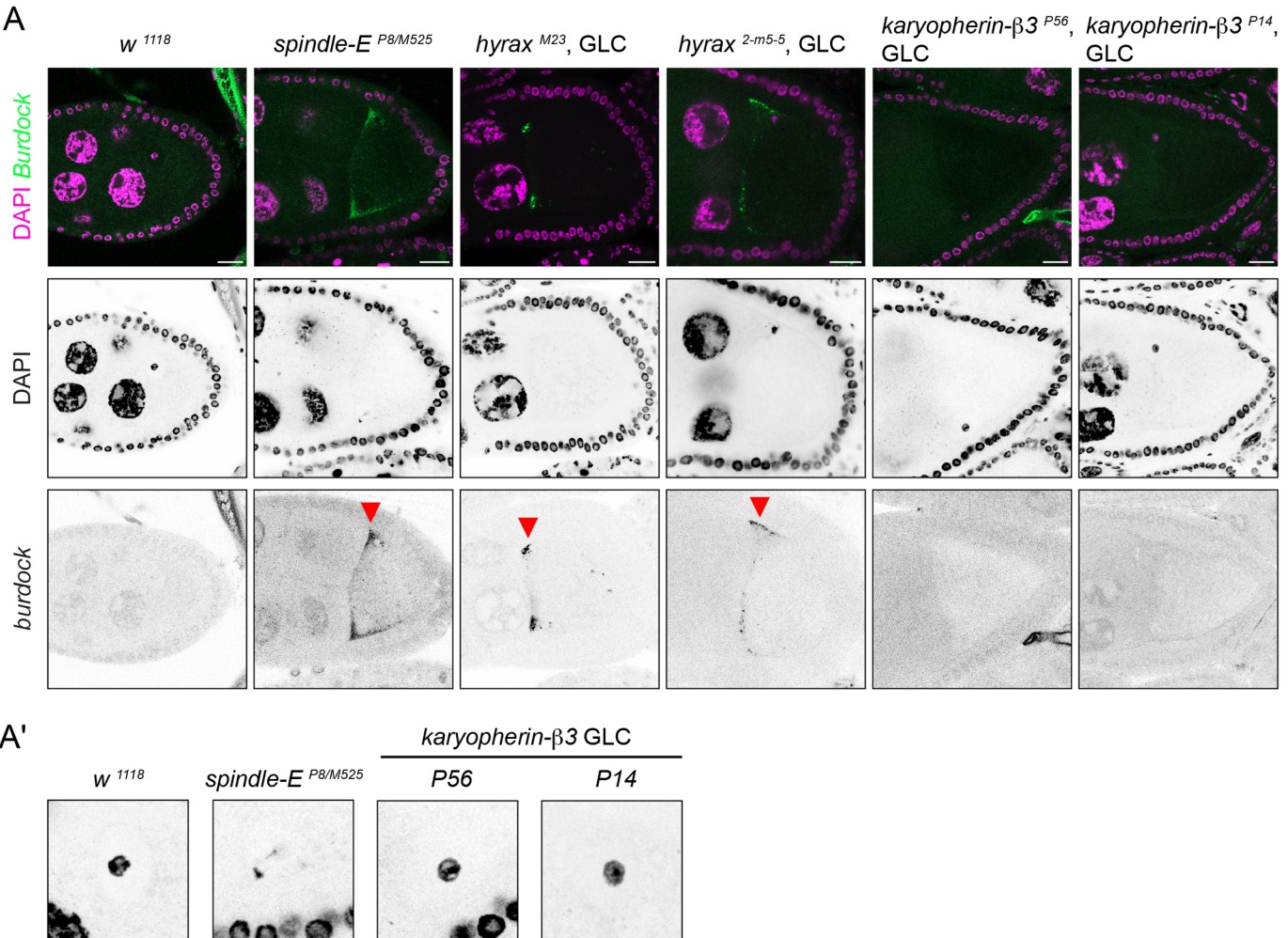

**Fig. 2. Novel mutations affecting *gurken* mRNA localisation.** (A) Shown are RNA fluorescent *in situ* hybridisation against *Burdock* transposon mRNA in the egg chambers of indicated genotypes. GLC, germline clones. Overexpressed *Burdock* mRNA is typically localised to the anterior end of the oocyte when it is expressed (marked by arrowheads). (A′) Shown are magnified views from panel A for the oocyte nuclear DNA stained by DAPI. *karyopherin-β3* mutant clones do not show the karyosome defect, indicating the absence of DNA damage. Scale bars: 20 µm.

(Fig. 5A, corresponding to stage P6 to P8), and the second egg chamber is clearly visible by stage P15 (100 APF) (Reilein et al., 2021). We found that none of the *armitage* mutant P11 ovaries carried intact egg chambers (Fig. 5B). Instead, the somatic cells failed to surround the germline cells, and the germline cells were often isolated and appeared to be undergoing cell death (Fig. 5B,B′). The phenotype was less severe in *vreteno* mutant pupae where intact egg chambers could be observed. However, germline cells were frequently lost from the germarium before progressing to make proper egg chambers (Fig. 5C).

The formation of one egg chamber instructs the formation of another that follows (Torres et al., 2003). We reasoned that the failure of the first couple of egg chambers in the piRNA pathway mutants might have caused the failure of all the other egg chambers that followed. Therefore, we focused on stage P6 to P8 when the basal stalk forms to hold the first egg chamber. The basal stalk is derived from a group of somatic cells that are posterior to the germline cells and undergo cell-cell intercalation to form a single columnar structure (Reilein et al., 2021) (Fig. 5A). The basal stalk precursor cells become flattened as they converge and make contacts at the midline of the future stalk as evidenced by foci of the cell-cell adhesion molecule DE-Cadherin, indicative of a

coordinated cell-cell intercalation (Fig. 5D). In contrast, basal stalk precursor cells in the piRNA pathway mutants failed to make tight contacts or made contacts in a semi random fashion as shown by discontinuous DE-Cadherin accumulation (Fig. 5D′,D″). This phenotype was consistently observed across many mutant ovarioles (Fig. S2). Interestingly, a similar phenotype was observed in *piwi* mutant larval ovaries where the somatic cells failed to intermingle with the germline cells (Saito et al., 2009). Although the precise mechanism remains unclear, the piRNA pathway and the silencing of *gypsy* retrotransposons may be generally required for coordinated movement of ovarian somatic cells.

In contrast to the whole mutant ovaries, RNAi against somatic piRNA pathway genes, including *armitage* and *vreteno*, using the pan-ovarian somatic driver *TJ-gal4* did not affect the gross morphology of egg chambers (Handler et al., 2011, 2013). Interestingly, we found that *armitage* and *piwi* are expressed in all basal stalk precursor cells, including those that express very little TJ (Fig. 5E,E′), likely explaining the discrepancy of the egg follicle phenotype. Basal stalk precursor cells, though initially far away from the germline cells, eventually integrate into the epithelium of the first couple of egg chambers (Reilein et al., 2021). These observations suggest that *gypsy* retrotransposons inhabit the basal stalk precursor cells for later

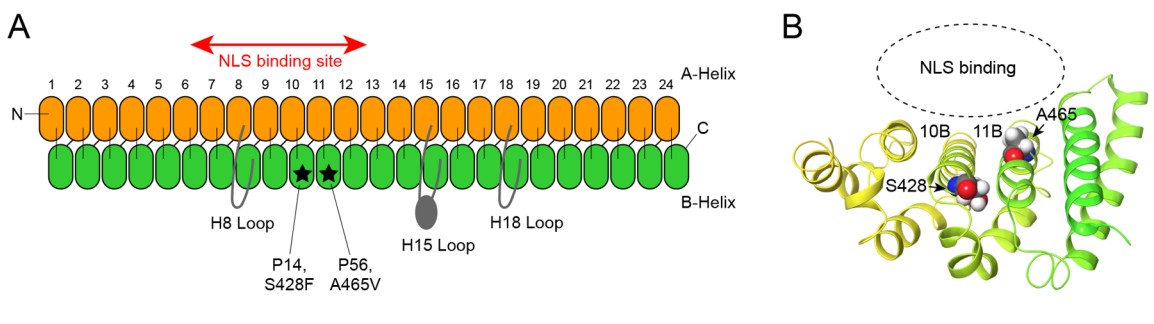

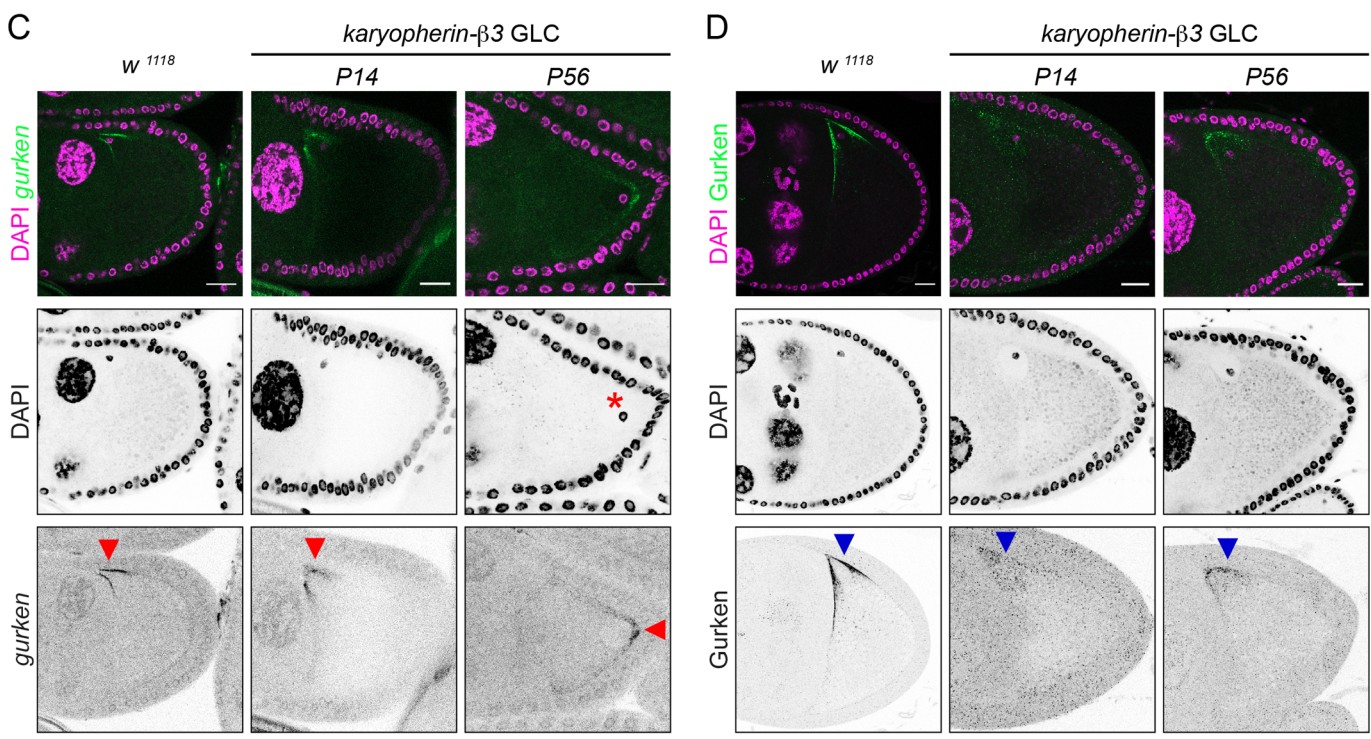

**Fig. 3. *karyopherin-β3* is required for efficient localisation and translation of *gurken* mRNA.** (A) Domain architecture of Karyopherin-β3/Kap121p, modified from the previous study (Kobayashi and Matsuura, 2013). Orange and green ovals indicate the A- and B-Helices of the HEAT repeat. Stars indicate the sites of mutations found in the screen. (B) Alphafold-predicted structure of the 10th and 11th HEAT repeat of Karyopherin-b3. The binding pocket of nuclear localisation signals (NLSs) and the positions of mutated amino acids are highlighted. (C,D) Shown are RNA fluorescent *in situ* hybridisation against *gurken* mRNA in C and immuno-staining against Gurken protein in D. *gurken* mRNA is tightly localised to the dorso-anterior corner of the mid-stage oocyte (indicated by red arrowheads) where the nucleus is also found and produces TGF-α like ligand protein Gurken (indicated by blue arrowheads). In the *karyopherin-β3* mutant oocytes, *gurken* mRNA is not tightly localised to the dorso-anterior corner and only weakly translated. The oocyte nucleus is always found at the anterior end of wild-type egg chambers, while it is frequently mis-localised to the posterior end of the *karyopherin-β3* mutant oocytes (shown by asterisk), indicating reduced Gurken activity in the earlier stages of oogenesis. Scale bars: 20 μm.

infection of the germline cells, and that the piRNA pathway protects ovaries against them. The failure of basal stalk formation may benefit the host by eliminating a lineage of egg chambers that are under threat of *gypsy* infection. It is an intriguing question for future studies which *gypsy* retrotransposons occupy the basal stalk and if this is an evolutionarily conserved mechanism (Senti et al., 2025).

### The Zinc Finger motif of Spindle-E is required for robust ping-pong biogenesis and transposon silencing

Of the six *spindle-E* mutations from the screen, five mutations showed the strong phenotype of *spindle-E* where females transheterozygous with the null allele *spindle-E¹* laid very few eggs with severely distorted morphology. The strength of these five mutations was indistinguishable from the *spindle-E¹* allele. In contrast, the *P64* allele showed a hypomorphic phenotype and females transheterozygous with *spindle-E¹* laid eggs with few

morphological defects despite being fully sterile. Sequencing identified a tyrosine missense mutation of cysteine 1400 (Table 3) in the *P64* chromosome. Cys1400 is found in the Zinc Finger motif of Spindle-E, which is present in the fly homologue but absent in the mouse homologue TDRD9 (Fig. 6A). The fly and mouse homologues share the rest of the protein, consisting of a Tudor domain and an RRM domain inserted into the extended DExH box RNA helicase domain that resembles DHX36 (Chen et al., 2018) (Fig. 6A; Fig. S3). The Zinc Finger is universally conserved in metazoans (Fig. 6B). Thus, it is a specific loss of the Zinc Finger in the mouse *TDRD9* gene. Interestingly, Ott et al. reported another mutation in the Zinc Finger motif, which also displayed the hypomorphic phenotype (Ott et al., 2014) (Fig. 6B). Intrigued by these findings, we decided to further investigate the phenotype of the *P64* Zinc Finger mutant, using females transheterozygous for the Zinc Finger and the null mutations (*P8* and *P64*, referred to as

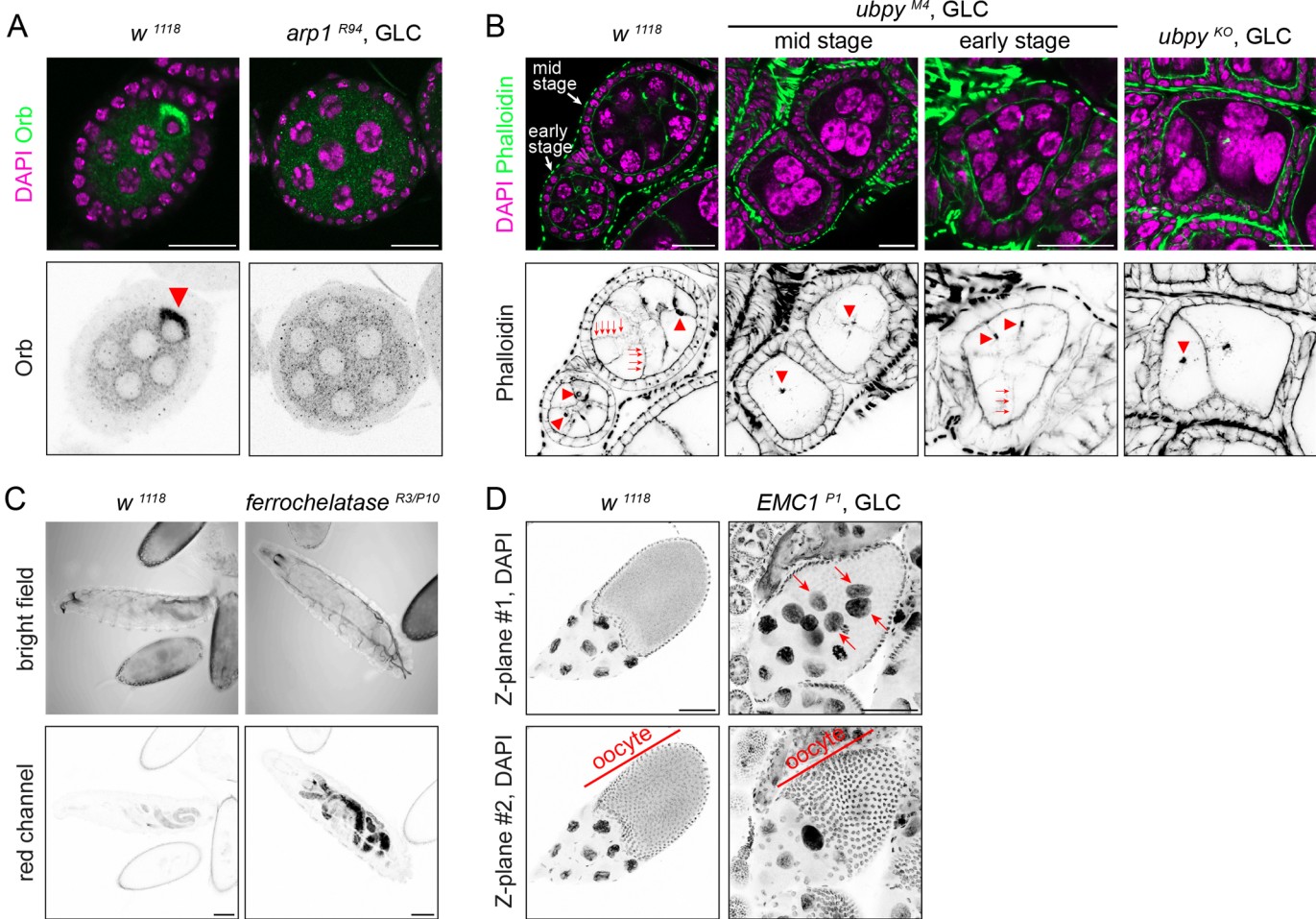

**Fig. 4. Miscellaneous phenotypes found in the screen.** (A) Immuno-staining against the oocyte marker Orb of the early-stage egg chambers of indicated genotypes. GLC, germline clones. One germline cell is selected to become the oocyte (indicated by an arrowhead) in wild-type egg chambers, while there is no asymmetric distribution of Orb in the *arp1* mutant egg chambers, indicating a failure of oocyte specification. (B) Shown are early to mid-stage egg chambers of indicated genotypes stained by the F-actin marker phalloidin. Nurse cell plasma membranes (marked by arrows) become invisible from mid-stage oogenesis in the *ubpy* mutants, suggesting a cell membrane rupture. Concomitantly, ring canal structure (marked by arrowheads) appears abnormal in the mutant and the number of ring canals per egg chamber decreases, suggesting that they coalesce. (C) Bright field and red channel images of the 1st instar larvae of indicated genotypes are shown. Substances that fluoresce red accumulate in the guts and other organs of *ferrochelatase* mutant larvae. (D) Shown are the late-stage egg chambers stained by DAPI. Nurse cell nuclei remain contained in the nurse cells as the oocyte grows its volume in the wild-type egg chambers while a few nurse cell nuclei (marked by arrows) intrude into the oocyte of the *EMC1* mutant egg chambers, indicating a defect in holding the nuclei during nurse cell dumping. Scale bars: 20 µm in A and B, and 100 µm in C and D.

*ZF/null*), and for the two different null mutations (*P8* and *P53*, referred to as *null/null*).

Consistent with the hypomorphic egg phenotype, RNA sequencing of the *spindle-E* Zinc Finger mutant ovaries showed a weaker transposon overexpression phenotype compared to the null *spindle-E* mutant ovaries (Fig. 6C). While telomeric transposons such as *HeT-A* and *TAHRE* were similarly overexpressed in both mutant combinations, other germline transposons such as *Burdock*, *Max-element*, *invader2*, *gypsy12* and *GATE* were either only mildly overexpressed or not overexpressed in the Zinc Finger mutant ovaries (Fig. 6C). The weak transposon overexpression phenotype corresponded to a mild impact on piRNA production. Small RNA sequencing revealed a marked reduction in the abundance of anti-sense piRNAs against most germline transposons in the *spindle-E* mull mutant ovaries (10- to 100-fold reduction compared to wild-type ovaries). On the other hand, the effects were milder in the Zinc Finger mutant ovaries where no transposon piRNAs, neither from anti-sense nor sense strands, were impacted more than 10-fold

(Fig. 6D; Fig. S4A,B). Spindle-E is essential for ping-pong piRNA biogenesis in flies and silkworms (Malone et al., 2009; Nishida et al., 2015). Interestingly, we found that ping-pong piRNA biogenesis was severely impaired in the Zinc Finger mutants, although it is not completely abolished compared to the *spindle-E* null mutants (Fig. 6E; Fig. S4C,D). On the other hand, the phasing mechanism, the alternative route for piRNA biogenesis, did not appear to be affected in the Zinc Finger mutants, explaining why overall piRNA abundance was maintained (Fig. S4E).

Ping-pong biogenesis operates in the peri-nuclear body called nuage through reciprocal endonucleolytic cleavage mediated by the two Argonaute proteins, Aubergine and AGO3. Localisation of Aubergine and AGO3 to nuage is an indicator of robust ping-pong biogenesis (Lim and Kai, 2007). *spindle-E* null mutations almost completely abolished the nuage localisation of Aubergine and AGO3, consistent with the loss of ping-pong and the previous study (Lim and Kai, 2007) (Fig. 6F,G). In contrast, the Zinc Finger mutation specifically affected the nuage localisation of AGO3, but

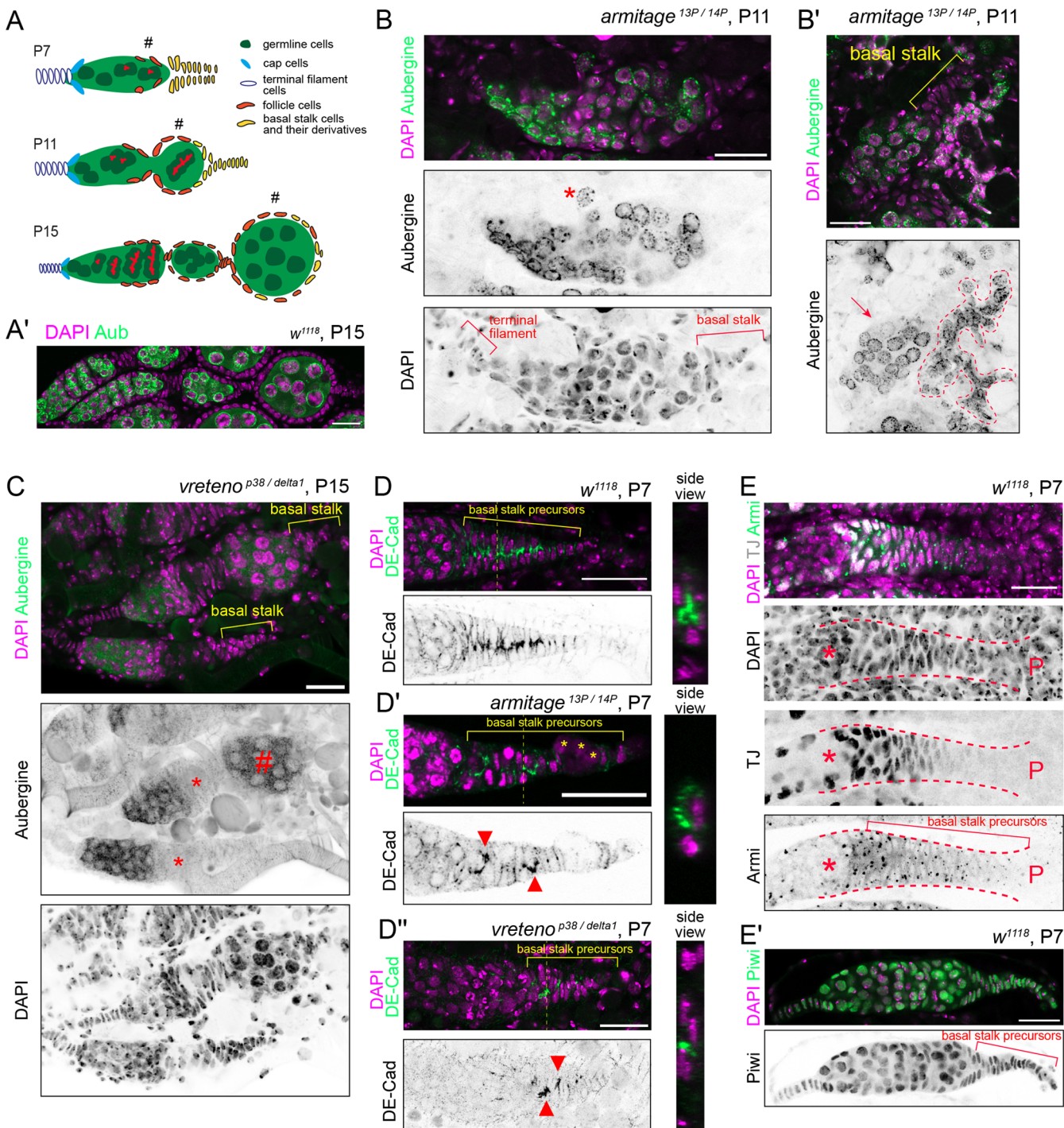

**Fig. 5. Loss of ovarian somatic piRNAs leads to defective basal stalk formation and failure of the first egg chamber formation.** (A) Shown are schematics of ovarioles from different pupal stages with different cell types coloured differently. Only the follicle cells that reside at the posterior half of the ovarioles are depicted. The first egg chambers are marked by hash tags. Basal stalks form between stages P7 and P11. A subpopulation of the basal stalk becomes part of the follicle epithelium of the first egg chambers (Reilein et al., 2021). (A′) A stage P15 wild-type ovariole, stained for the germline marker Aubergine, consisting of the germarium and the first couple of egg chambers. (B,B′) Shown are stage P11 *armitage*[13P/14P] mutant ovarioles, stained for Aubergine. The germline cells are frequently detached from the germarium (marked by asterisk and a dashed line). The germline cyst that is held by the basal stalk is indicated by an arrow. (C) Shown is an overlaid Z-stack image of P15 *vreteno*[p38/delta1] mutant ovarioles, showing the absence of germline cells (asterisk). Twenty-three Z-stack images, each spaced 0.26 mm apart, were merged to cover the entire ovarioles and demonstrate the absence of germline cells. The germline cyst that remains covered by the somatic cells is marked by a hash tag. (D-D″) Shown are stage P7 ovarioles of indicated genotypes stained for DE-Cadherin. The representative image of the Z-stack and the side view of the compound images are shown to the left and right, respectively. There is a strong continuous signal of DE-Cadherin between intercalating basal stalk precursor cells in the wild-type ovariole, while strong signals of DE-Cadherin, marked by arrowheads, are not continuous in the *armitage* and *vreteno* mutant ovarioles. Cells of unknown origin often attach to the basal stalk of the *armitage* mutant ovarioles (asterisk). (E,E′) Staining of Armitage and Piwi in stage P7 wild-type ovarioles, showing their expression in the basal stalk precursor cells. Traffic Jam (TJ) is strongly expressed in the somatic cells near and around the germline cells (asterisk) while it is not expressed in the basal stalk precursor cells near the posterior end (marked by 'P'). Scale bars: 20 μm.

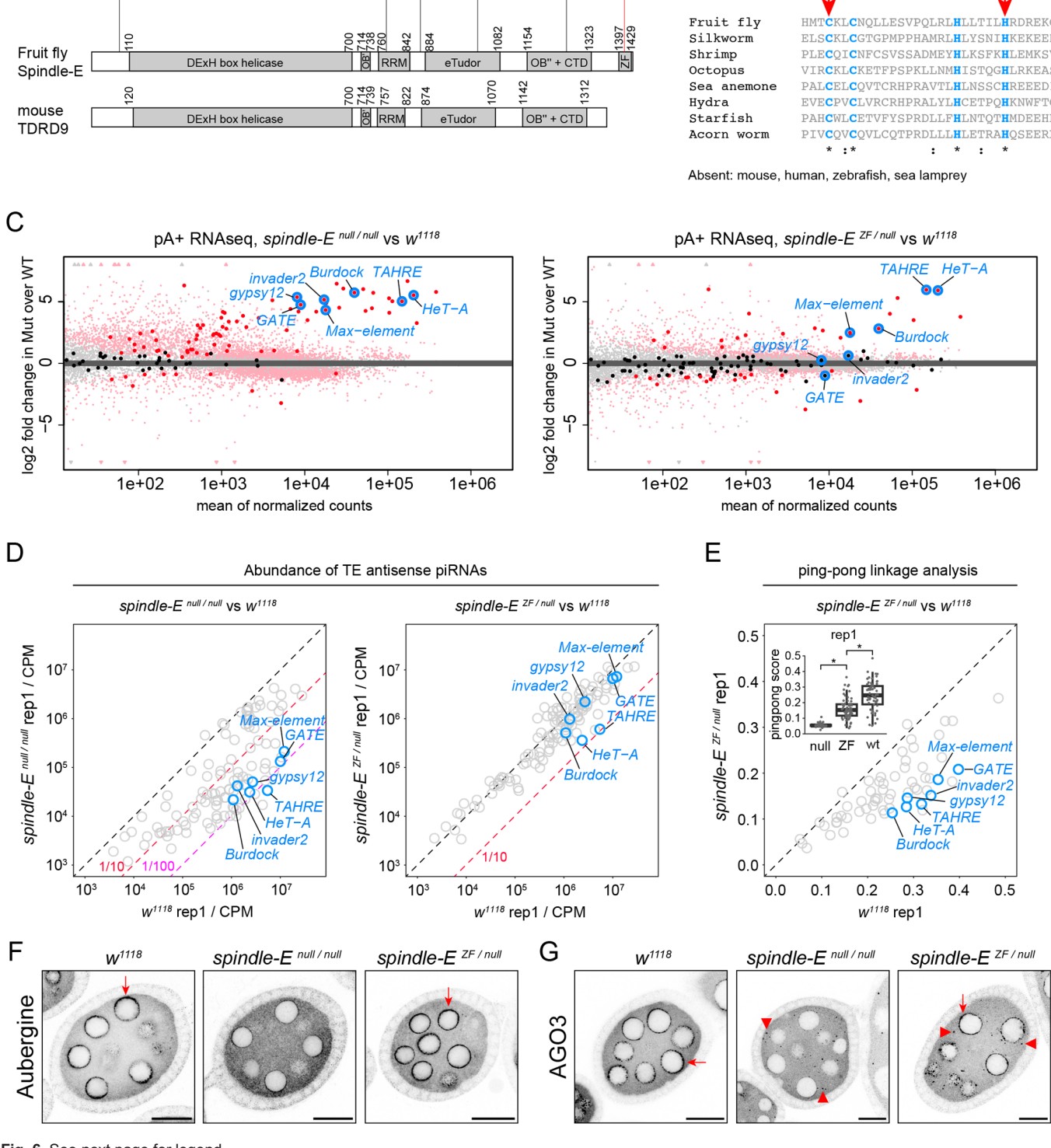

**Fig. 6.** See next page for legend.

not of Aubergine, with some AGO3 proteins forming small granules in the cytoplasm away from the nuclear envelope (Fig. 6F,G). Neither mutation qualitatively affected the nuclear localisation of Piwi, which is mainly loaded with piRNAs made by phasing (Fig. S5).

The ping-pong and phasing piRNA biogenesis mechanisms are mechanistically and physically connected where the slicing of an RNA by Aubergine/AGO3 within the ping-pong cycle can also

initiate phasing biogenesis (Han et al., 2015; Mohn et al., 2015). This is mediated through the RNA helicase Armitage that shuttles between nuage and mitochondria, where phasing occurs through endonucleolytic cleavage by the membrane protein Zucchini/MitoPLD (Saito et al., 2010; Olivieri et al., 2010; Ge et al., 2019). Interestingly, the AGO3-piRNA complex is more potent in triggering phasing than the Aubergine-piRNA complex (Mohn et al., 2015),

**Fig. 6. The Zinc Finger motif of Spindle-E is required for efficient ping-pong piRNA biogenesis and transposon silencing.** (A) Domain architecture of fruit fly Spindle-E and mouse TDRD9. Positions of the six mutations identified in the screen are shown. Hash tag indicates a splice site mutation (see Table 3). RRM and extended Tudor (eTudor) domains are inserted in the middle of the predicted OB-fold (denoted as OB′ and OB″). See also Fig. S2. (B) Alignment of predicted Zinc Finger motifs in Spindle-E homologues and absence thereof. Cysteine and histidine residues that make up the C2H2-type Zinc Finger are coloured in cyan. Sequences examined are: fruit fly (*Drosophila melanogaster*, FBpp0082637), silkworm (*Bombyx mori*, XP_012544435.2), shrimp (*Artemia franciscana*, XP_065584778.1), octopus (*Octopus bimaculoides*, XP_014767807.1), sea anemone (*Nematostella vectensis*, XP_048586381.1), hydra (*Hydra vulgaris*, XP_047133149.1), starfish (*Acanthaster planci*, XP_022106993.1), acorn worm (*Ptychodera flava*, XP_070576791.1), mouse (XP_006516379.1), human (XP_054231303.1), zebrafish (XP_068068947.1), see lamprey (*Petromyzon marinus*, XP_032811802.1) (C) MA plots made by DESeq2 showing the differential expression of *Drosophila* genes (small dots) and transposons (large dots) in the ovaries. The plots compare triplicates of the wild-type (*w1118*), the Zinc Finger mutant (ZF) and the null mutants of *spindle-E*. Genes and transposons for which expression is significantly altered are coloured in red. Seven representative germline transposons are marked for a comparison. (D) Scatter plots showing the abundance of piRNAs (as defined by reads longer than 22 nucleotides) mapping to antisense transposon sequences. The plots compare replicate 1 of the wild-type (*w1118*), the Zinc Finger (ZF) and the null mutants of *spindle-E*. Replicate 2 is presented in Fig. S3A,B. The same seven representative transposons as panel C are highlighted. (E) Scatter plot comparing the ping-pong linkage values of transposon mapping piRNAs between the wild-type and the *spindle-E* Zinc Finger mutant. *Student's *t*-test $P < 1.0 \times 10^{-15}$. (F,G) Shown are early-stage egg chambers of indicated genotypes stained for Aubergine in F and AGO3 in G. Both Aubergine and AGO3 are tightly localised to the peri-nuclear body called nuage (arrows) in the wild-type egg chambers. Some AGO3 proteins are detached from the nuclear envelope and form granules (arrowheads) in the *spindle-E* Zinc Finger mutant egg chamber. Similar granules are also seen in the *spindle-E* null mutant egg chamber. Scale bars: 20 μm.

suggesting that AGO3 and Aubergine have different affinities to other components of the phasing process. A recent study in silkworm also revealed that Spindle-E specifically binds AGO3 to help release the cleavage product of the AGO3-piRNA complex (Izumi et al., 2024). This, combined with the differential affinity to the phasing machinery, likely explains the differential impacts on AGO3 and Aubergine by the *spindle-E* Zinc Finger mutation. The Spindle-E Zinc Finger motif is a C2H2-type that is most frequently found in DNA binding transcription factors but is also found in several double-stranded RNA binding proteins (Yang et al., 1999; Méndez-Vidal et al., 2002; Mackeh et al., 2018) (Fig. 6B). Double-stranded RNA binding proteins are commonly required for Dicer-dependent siRNA and microRNA synthesis in plants and animals (Liu et al., 2003; Lee et al., 2006; Curtin et al., 2008). Therefore, sensing of double-stranded RNA may be a common critical step for ancient small RNA biogenesis.

The 3R and 3L mutagenesis screens individually identified novel mutations affecting *gurken* mRNA localisation and other aspects of *Drosophila* oogenesis (Hayashi et al., 2014). They also reinforced each other's findings. For example, the *Tsc1* (3R) and *Tsc2* (3L) mutations both showed a germline overgrowth phenotype. Mutations in *EMC1* (3R) and *Chd5* (3L) genes, both of which are involved in the insertion of Tail-anchored proteins to the ER membrane, showed nurse cell dumping defects. Finally, the *vreteno* (3R) and *armitage* (3L) mutations commonly affected basal stalk formation. Furthermore, random mutagenesis uniquely revealed the function of the Spindle-E Zinc Finger, which was corroborated by the previous study also based on a mutagenesis screen (Ott et al., 2014). Thus, the mutagenesis-based forward genetic screen remains a powerful genetic tool for uncovering novel biological processes.

## MATERIALS AND METHODS

### Genetic screen and fly husbandry

The screen was performed as previously described (Hayashi et al., 2014) with modifications to allow generation of the germline clone of chromosome 3R. Briefly, *FRT82B* males were fed with EMS before being crossed with balancer females, from which we obtained females with a mutagenised third chromosome over the balancer chromosome. These mutagenised females were individually crossed to the tester males that carry *hsFLP*, *nanos::MCP-mCherry* (Weil et al., 2012), *gurken(MS2)12* (Weil et al., 2012), and *FRT82B P{ovoD1}* (Bloomington #2149). We heat shocked larvae to induce the generation of mitotic clones and dissected the F1 females that were transheterozygous for *FRT82B \** ("*FRT82A \**" representing the mutagenised chromosome 3R) and *FRT82B P{ovoD1}* 8-10 days after the heat shock. We screened for the distribution of fluorescently marked *gurken* mRNA in stage 8-9 oocytes. The dominant female-sterile mutation *ovoD1* was used to only allow the growth of mutant mitotic germline clones beyond stage 2-4.

We collected the F1 males from the vials, from which we collected the F1 females for the phenotyping, and crossed them with the balancer females to establish stocks while also immediately setting up a retest. We kept the stock when a phenotype repeated twice. We established complementation groups based on lethality or sterility over known mutations as well as between the mutated chromosomes. Mutant fly strains used are *vreteno^delta1* (gift from Julius Brennecke; Handler et al., 2011), *hyrax1* (gift from Konrad Basler; Mosimann et al., 2006), *Df(3R)BSC616* (for *kotsubu*, Bloomington #25691), *spindle-A1* (Bloomington #3322), *spindle-F1* (Bloomington #3325), *spindle-E1* (Bloomington #3327), *squid1* (gift from Ilan Davis), *ubpy^ko* (gift from Jin Jiang; Mukai et al., 2010), *Df(3R)BSC848* (for *bag of marbles*, Bloomington #29027), and *Tsc-1^Q87X* (gift from Nic Tapon). Novel mutations were identified by Sanger sequencing or the Illumina whole-genome sequencing conducted by the sequencing facility in Cancer Research UK using the genomic DNA extracted from homozygous 1st instar larvae. The identification of homozygous larvae was aided by the Tb marker on the TM6 balancer chromosome. The *hyrax^2-m5-5* mutant was made using the CRISPR guide RNA GCGGATTTCAGCGACATACCG with the first 'G' not being complementary to the genomic sequence. Flies were kept at room temperature (around 22°) or at 25° in standard food based on cornmeal or semolina supplemented with molasses, sugar, and fresh yeast.

### Immuno-staining

The following primary antibodies were used: anti Orb (mouse, DSHB 4H8, 1:100), anti-Gurken (mouse, DSHB 1D12, 1:100, anti-DE-Cadherin (rat, DSHB DCad2, 1:300), anti-Armitage (mouse monoclonal 1D1-3H10 using the peptide from 36 to 230 aa of FBpp0100102, gift from Julius Brennecke, 1:500), anti-Traffic Jam (guinea pig, gift from Dorothea Godt, 1:20,000), anti-Piwi (mouse monoclonal 8C2-E4, gift from Julius Brennecke, 1:500), anti-Aubergine (mouse monoclonal 8A3-D7, gift from Julius Brennecke, 1:500) and anti-AGO3 (mouse monoclonal 7B4-C2, gift from Julius Brennecke, 1:500). The ovaries were freshly dissected from 2- to 7-day-old females in PBS and fixed in PBS containing 4% formaldehyde for 10 min at room temperature. Fixed ovaries were permeabilised in PBS containing 0.5% v/v Triton-X for 30 min, washed in PBS containing 0.1% Triton-X (PBS-Tx) several times before blocking in PBS-Tx containing 0.05% w/v bovine serum albumin for 30 min. The primary antibody incubation was conducted in the blocking solution at 4°C overnight. Goat anti-rabbit, mouse, rat and guinea pig IgG conjugated to Alexa Fluor 488, 568 or 647 (Abcam) were used as secondary antibodies, and the confocal images were taken on Zeiss LSM-800 and LSM-780 at the Centre for Advanced Microscopy at the ANU. Images were processed by Fiji.

### RNA fluorescent *in situ* hybridisation

RNA fluorescent *in situ* hybridisation against *Burdock* and *gurken* mRNAs were done as previously described (Chary and Hayashi, 2023) using the short oligo DNAs for each target (Table S1).

### Small RNA sequencing

Oxidised small RNA libraries were prepared as previously described (Chary and Hayashi, 2023). Briefly, approximately ten pairs of ovaries were

dissected from 2- to 7-day-old females. Ovaries were transferred to TRIZOL (Thermo Fisher Scientific) for RNA extraction followed by DNase I (Thermo Fisher Scientific) digest to remove genomic DNA. We generated small RNA libraries from 1-5 µg of total RNA using a modified protocol from the original method (Hafner et al., 2008). 19- to 35-nucleotides-long RNA was first selected by polyacrylamide gel electrophoresis (PAGE) with 7 M Ureal using radio-labelled 19mer (5′-CGUACGCGGGUUUAAACGA) and 35mer spikes (5′-CUCAUCUUGGUCGUACGCGGAAUAGUUU AAACUGU). The size-selected RNA was precipitated, oxidised by sodium periodate (Sigma-Aldrich) (Akbergenov, 2006), and size-selected for the second time by PAGE. The size-selected oxidised small RNAs were then ligated to the 3′ adapter from IDT (5rApp/NNNNAGATCGGAAGAG CACACGTCT/3ddC where Ns are randomised) using the truncated T4 RNA Ligase 2, truncated KQ (NEB), followed by a third PAGE to remove non-ligated adapters. Subsequently, the RNA was ligated to the 5' adapter from IDT (5′-ACACUCUUUCCCUACACGACGCUCUUCCGAUCU NNNN where Ns are randomised) using the T4 RNA Ligase 1 (NEB). Adapter-ligated RNA was reverse-transcribed using SuperScript II (Thermo Fisher Scientific) and amplified by KAPA LongRange DNA polymerase (Sigma-Aldrich) using the universal forward primer, Solexa_PCR-fw: (5′-AATGATACGGCGACCACCGAGATCTACACTCTTTCCCTACACG-ACGCTCTTCCGATCT) and the barcode-containing reverse primer TruSeq_IDX: (5′-CAAGCAGAAGACGGCATACGAGATyyyyyyGT GACTGGAGTTCAGACGTGTGCTCTTCCGATCT where yyyyyy is the reverse-complemented barcode sequence). Amplified libraries were multiplexed and sequenced on an Illumina Novaseq platform in the paired-end 150 bp mode by GENEWIZ/Azenta. Processed sequencing data for figure generation are available in Table S2.

### Small RNA sequencing analysis

The R1 sequencing reads were trimmed of the Illumina-adapter sequence AGATCGGAAGAGCACACGTCTGAACTCCAGTCAC using the FASTX-Toolkit from the Hannon Laboratory (CRUK Cambridge Institute). The four random nucleotides at either end of the read were further removed. The trimmed reads of 18 to 40 nucleotides in size were first mapped to the infrastructural RNAs, including ribosomal RNAs, small nucleolar RNAs (snRNAs), small nuclear RNAs (snoRNAs), microRNAs, and transfer RNAs (tRNAs) using Bowtie 1.2.3 allowing up to one mismatch. Sequences annotated in the dm6 r6.31 assembly of the *Drosophila melanogaster* genome were used. The trimmed and unfiltered reads were mapped to the curated sequences of *Drosophila melanogaster* transposons (Senti et al., 2015) using Bowtie allowing up to three mismatches with the option of –all –best –strata. Bedtools 2.28.0 was used to count the coverage of the mapped reads. Endogenous siRNA reads (annotated as hpRNA) were used for normalisation. Ping-pong linkage score was measured as previously described (Chary and Hayashi, 2023). Nucleotide compositions around the 3′ ends of the antisense transposon-mapping piRNA reads (longer than 22 nucleotides) were counted and visualised using weblogo 3.7.8. Frequencies were measured in the window of 11 nucleotides and the z scores were calculated as the deviation of the frequency value at the immediate downstream position of piRNA 3′ end (+1 position) from the mean frequency divided by the standard deviation of the frequencies.

### RNA sequencing and analysis

Polyadenylated RNA was purified from the DNase-treated total RNA using oligo d(T)25 magnetic beads (NEB) and used for library preparation. Libraries were cloned using the NEBNext Ultra Directional II RNA Library Prep Kit for Illumina (NEB), following the manufacturer's instruction, amplified by KAPA polymerase using the same primers as for the small RNA sequencing, and sequenced using an Illumina Novaseq platoform in the paired-end 150 bp mode. Both R1 and R2 reads from the polyA-selected RNA sequencing reads were trimmed of the Illumina-adapter sequences using the FASTX-Toolkit. The trimmed reads were subsequently filtered by the sequencing quality. Only the paired and unfiltered reads were then mapped to the dm6 r6.31 transcriptome combined with curated sequences of *Drosophila melanogaster* transposons using salmon/1.1.0 with the options of –-validateMappings –-incompatPrior 0.0 –-seqBias –-gcBias. The transcript counts output files of salmon (quant.sf) were imported to DESEqn (1.42.0;

Love et al., 2014) to perform differential gene expression analysis. Processed sequencing data for figure generation are available in Table S2.

### Structure prediction of Karyopherin-β3 and Spindle-E/TDRD9

We used the colabfold wrap v1.4.0 of Alphafold 2 (https://github.com/YoshitakaMo/localcolabfold/releases/tag/v1.4.0), and used MMSeqs2 for the database search and clustering of protein sequences before performing the structural prediction (Mirdita et al., 2022). Sequences used for the analysis are FBpp0078500 for Karyopherin-b3, FBpp0082637 for Spindle-E and XP_006516379.1 for mouse TDRD9. Predicted structures were analysed using Maestro (Schrödinger Release 2025-1).

#### Acknowledgements
We thank the sequencing facility at Cancer Research UK and Stuart Horswell for sequencing the fly genomes and identifying mutations. We thank Peter Duchek and Joseph Gokcezade (IMBA fly facility) for generating CRISPR-edited *hyrax*[2-m5-5] mutant flies during R.H.'s affiliation with IMBA. We thank Julius Brennecke, Konrad Basler, Ilan Davis, Jin Jiang, Nic Tapon and Dorothea Godt for providing fly mutant strains and antibodies. We thank the fly community in the London Research Institute (Tapon and Thompson groups) for valuable suggestions and discussion for the mutant phenotype characterisation.

#### Competing interests
The authors declare no competing or financial interests.

#### Author contributions
Conceptualization: D.I.-H., R.H.; Data curation: S.J.L.; Formal analysis: S.J.L., R.H.; Funding acquisition: D.I.-H., R.H.; Investigation: S.J.L., R.T., J.-S.L., S.S., S.C., A.H., S.M.W., S.P., R.H.; Methodology: S.J.L.; Project administration: D.I.-H.; Resources: R.H.; Supervision: D.I.-H., R.H.; Writing – original draft: R.H.; Writing – review & editing: R.H.

#### Funding
This work was initially supported by Cancer Research UK (to D.I.H.) and then by the Australian Research Council (DP210102385 to R.H.). Open Access funding provided by Australian National University. Deposited in PMC for immediate release.

#### Data and resource availability
Sequencing data and processed files have been deposited to Gene Expression Omnibus (GSE293862 and GSE293864). Code for the analysis of the poly-A selected RNA sequencing and the small RNA sequencing data is available through github (https://github.com/RippeiHayashi/3R_screen/tree/main). All other relevant data and details of resources can be found within the article and its supplementary information.

#### Peer review history
The peer review history is available online at https://journals.biologists.com/bio/lookup/doi/10.1242/bio.062321.reviewer-comments.pdf

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
