## [Peer Review File · Biology Open]

A second genetic screen for gurken mRNA mislocalisation uncovers novel phenotypes of piRNA pathway mutants in *Drosophila*

Sophie J. Liddell, Rahnya Taghi, Jessie-Siling Li, Sejal Sathe, Shashank Chary, Azusa Hayashi, S. Mark Wainwright, Sheena Pinchin, David Ish-Horowicz and Rippei Hayashi
DOI: 10.1242/bio.062321

Editor: Tristan Rodríguez

Review timeline

Original submission to sister journal:	16 April 2025
Editorial decision at sister journal:	22 May 2025
Transfer to Biology Open:	14 October 2025
Accepted:	15 October 2025

Original submission

First decision letter

MS Title: A second genetic screen for gurken mRNA mislocalisation uncovers novel phenotypes of piRNA pathway mutants in *Drosophila*

Authors: Rippei Hayashi; Sophie J. Liddell; Rahnya Taghi; Jessie-Siling Li; Sejal Sathe; Shashank Chary; Azusa Hayashi; S. Mark Wainwright; Sheena Pinchin; David Ish-Horowicz
Article Type: Research Article

I have now received all the referees' reports on the above manuscript and have reached a decision. I am sorry to say that the outcome is not a positive one. The referees' comments are appended below, or you can access them online: please go to .

As you will see, the referees raise some significant concerns about your paper, and are not strongly in favour of publication. Having looked at the manuscript myself, I agree with their views, and I must therefore, reject your paper.

I do realise this is disappointing news, but we receive many more papers than we can publish, and we can only accept manuscripts that receive strong support from referees.

I do hope you find the comments of the referees helpful, and that this decision will not dissuade you from considering our journal for publication of your future work. Many thanks for sending your manuscript to us.

Reviewer 1

SUMMARY OF THE ADVANCE MADE IN THIS PAPER AND ITS POTENTIAL SIGNIFICANCE TO THE FIELD

The manuscript from Hayashi et al reports the results of a genetic screen aimed at identifying genes required for gurken mRNA localization in the developing *Drosophila* oocyte. This screen was conducted over a decade ago by one of the co-authors during her PhD studies in the lab of David Ish-Horowicz, who sadly is now deceased. The authors deserve credit for taking the time and effort to assemble this manuscript, as often in such situations interesting data remain unpublished and are essentially lost.

The screen enabled the identification of novel alleles in several genes of the piRNA pathway that is already known to be essential for microtubule-dependent mRNA localization. These new alleles reveal a role for the piRNA pathway in basal stalk development. It also identified a new gene, *hyrax*, that functions in transposon silencing, and clarified the role of the zinc-finger domain of Spindle-E in ping-pong piRNA biogenesis. These are interesting observations that deserve publication.

However, none of these observations in my view has been sufficiently developed to constitute a significant and novel contribution to our understanding of developmental mechanisms. In several instances the authors identify important questions that if answered could potentially could make such a contribution, and in some cases they propose appropriate follow-up experiments to address them. But these experiments would take an extended period of time to do, beyond the timeframe that is reasonable for revisions to a paper. I therefore recommend that this manuscript is redirected to another more specialized journal.

Minor comments: the authors state that there is a strong focused localization of *gurken* mRNA in the *karyopherin-beta3* mutant but this is not at all clear to me from the images in Fig 3D. Also in Fig 3C and 3D the orientation of the egg chambers are not uniform, dorsal is up in the wild-type control but down in the mutant. This is confusing to the reader. Finally, the text would benefit from some light copyediting, as there are some odd turns of phrase. Examples are on line 39: 'transport occurs in two flavours' and on line 87-88, in that chromosomes don't show *gurken* mis-localization, oocytes do.

Reviewer 2

SUMMARY OF THE ADVANCE MADE IN THIS PAPER AND ITS POTENTIAL SIGNIFICANCE TO THE FIELD

This manuscript reports a follow-up to a previous EMS mutagenesis screen on chromosome arm 3L (Hayashi et al., 2014, G3), extending the strategy to chromosome 3R in *Drosophila* to identify factors involved in *gurken* mRNA localization during oogenesis. In the present study, the authors use a similar approach and uncover several genes, including new alleles in known piRNA pathway components as well as novel regulators such as *hyrax* and *karyopherin-β3*. They also characterize a Zinc Finger domain mutation in *spindle-E*, proposing functional implications for AGO3 localization and piRNA biogenesis.

While the genetic screen is carefully executed and technically sound, the mechanistic depth of the study is limited. For most of the newly identified factors, the analysis remains largely phenotypic, with minimal insight into the molecular mechanisms underlying the observed defects. The analysis of *spindle-E* is the most developed, yet even there, further evidence is needed to support the proposed model of domain-specific function in the ping-pong pathway. Given the descriptive nature of the findings and the lack of deeper mechanistic investigation, the manuscript does not currently meet the standards of conceptual advance typically expected for Development. A more targeted and mechanistically driven focus on one or two of the most promising observations could significantly strengthen the manuscript.

SUGGESTIONS TO AUTHORS

Major Concerns

1. The analysis of basal stalk defects in *vreteno* and *armitage* mutants is central to the model but is not quantified. The authors should provide data on the frequency and severity of stalk abnormalities across pupal stages (e.g., P6-P15), including penetrance values and representative images.
2. The conclusion that *hyrax* is essential for transposon silencing is currently supported only by *burdock* RNA in situ hybridization. Given that *hyrax* encodes a homolog of the PAF1 complex subunit, it would be important to clarify whether the observed phenotype results from impaired transcription of piRNA clusters or precursor RNAs. To further investigate the underlying cause,

additional evidence—such as qPCR of other transposons, RNA-seq, or piRNA-seq from hyrax germline clones—should be provided to assess whether the defect broadly affects piRNA biogenesis and transposon repression.

3. The Zinc Finger mutation of spindle-E is proposed to specifically disrupt AGO3 localization and function, but the mechanistic basis is underexplored. It would strengthen the argument to test AGO3-piRNA complex integrity or its interaction with partners like Krimper via immunoprecipitation and small RNA analysis.

First revision

Author response to reviewers' comments

Point-by-point response to the reviewers' comments (response in blue)

We thank the reviewers for providing valuable comments and suggestions to improve our manuscript. Below we outlined our response to their comments.

Reviewer 1: SUMMARY OF THE ADVANCE MADE IN THIS PAPER AND ITS POTENTIAL SIGNIFICANCE TO THE FIELD

The manuscript from Hayashi et al reports the results of a genetic screen aimed at identifying genes required for gurken mRNA localization in the developing Drosophila oocyte. This screen was conducted over a decade ago by one of the co-authors during her PhD studies in the lab of David Ish-Horowicz, who sadly is now deceased. The authors deserve credit for taking the time and effort to assemble this manuscript, as often in such situations interesting data remain unpublished and are essentially lost.

The screen enabled the identification of novel alleles in several genes of the piRNA pathway that is already known to be essential for microtubule-dependent mRNA localization. These new alleles reveal a role for the piRNA pathway in basal stalk development. It also identified a new gene, hyrax, that functions in transposon silencing, and clarified the role of the zinc-finger domain of Spindle-E in ping-pong piRNA biogenesis. These are interesting observations that deserve publication.

However, none of these observations in my view has been sufficiently developed to constitute a significant and novel contribution to our understanding of developmental mechanisms. In several instances the authors identify important questions that if answered could potentially make such a contribution, and in some cases they propose appropriate follow-up experiments to address them. But these experiments would take an extended period of time to do, beyond the timeframe that is reasonable for revisions to a paper. I therefore recommend that this manuscript is redirected to another more specialized journal.

Minor comments:

1. the authors state that there is a strong focused localization of gurken mRNA in the karyopherin-beta3 mutant but this is not at all clear to me from the images in Fig 3D.

We meant in the original text that a strong and focused localisation of Gurken proteins to the dorsal anterior corner of oocytes is seen in the wild-type as shown in Fig 3D.

2. Also in Fig 3C and 3D the orientation of the egg chambers are not uniform, dorsal is up in the wild-type control but down in the mutant. This is confusing to the reader.

We flipped images to keep the dorsal side consistently up in the egg chambers in the revised Figure 3.

3. Finally, the text would benefit from some light copyediting, as there are some odd turns of

phrase. Examples are on line 39: 'transport occurs in two flavours' and on line 87-88, in that chromosomes don't show gurken mis-localization, oocytes do.

We corrected the wording as suggested.

Reviewer 2: SUMMARY OF THE ADVANCE MADE IN THIS PAPER AND ITS POTENTIAL SIGNIFICANCE TO THE FIELD

This manuscript reports a follow-up to a previous EMS mutagenesis screen on chromosome arm 3L (Hayashi et al., 2014, G3), extending the strategy to chromosome 3R in Drosophila to identify factors involved in gurken mRNA localization during oogenesis. In the present study, the authors use a similar approach and uncover several genes, including new alleles in known piRNA pathway components as well as novel regulators such as hyrax and karyopherin-β3. They also characterize a Zinc Finger domain mutation in spindle-E, proposing functional implications for AGO3 localization and piRNA biogenesis.

While the genetic screen is carefully executed and technically sound, the mechanistic depth of the study is limited. For most of the newly identified factors, the analysis remains largely phenotypic, with minimal insight into the molecular mechanisms underlying the observed defects. The analysis of spindle-E is the most developed, yet even there, further evidence is needed to support the proposed model of domain-specific function in the ping-pong pathway. Given the descriptive nature of the findings and the lack of deeper mechanistic investigation, the manuscript does not currently meet the standards of conceptual advance typically expected for Development. A more targeted and mechanistically driven focus on one or two of the most promising observations could significantly strengthen the manuscript.

SUGGESTIONS TO AUTHORS

Major Concerns

1. *The analysis of basal stalk defects in vreteno and armitage mutants is central to the model but is not quantified. The authors should provide data on the frequency and severity of stalk abnormalities across pupal stages (e.g., P6-P15), including penetrance values and representative images.*

In the new Supplementary Figure S2, we provided six images of P7 ovarioles from the wild-type, vreteno and armitage mutant pupal ovaries, showing that the continuous DE-Cad staining in the basal stalk is consistently seen only in the wild-type ovarioles. We added a sentence in line 198 to emphasise the penetrance of the said phenotype. The numbering of the Supplementary Figures has been changed in the revised manuscript because of this new figure.

2. *The conclusion that hyrax is essential for transposon silencing is currently supported only by burdock RNA in situ hybridization. Given that hyrax encodes a homolog of the PAF1 complex subunit, it would be important to clarify whether the observed phenotype results from impaired transcription of piRNA clusters or precursor RNAs. To further investigate the underlying cause, additional evidence—such as qPCR of other transposons, RNA-seq, or piRNA-seq from hyrax germline clones—should be provided to assess whether the defect broadly affects piRNA biogenesis and transposon repression.*

We thank the reviewer's suggestion. We performed the small RNA sequencing to quantify piRNAs expressed in ovaries bearing the mutant germline clones of the hyrax mutant alleles, M23 and 2-m5-5. As shown in the figure presented in this response letter, piRNA abundance changed only mildly and not consistently in the mutants compared to the wild-type. The extent of changes is considerably small compared to other piRNA pathway mutants that are known to affect the transcription of the piRNA cluster. However, we are reluctant to conclude that hyrax is not involved in the piRNA cluster transcription based on this observation alone. Therefore, we decided not to mention the results. More mechanistic characterisation is required to better understand the function of hyrax in transposon silencing as stated in the manuscript.

3. *The Zinc Finger mutation of spindle-E is proposed to specifically disrupt AGO3 localization and function, but the mechanistic basis is underexplored. It would strengthen the argument to test AGO3-piRNA complex integrity or its interaction with partners like Krimper via immunoprecipitation and small RNA analysis.*

We agree with the reviewer that it is highly informative to assess the physical interaction between AGO3 and other piRNA biogenesis factors, and how it is affected in the *spindle-E* zinc finger mutant ovaries. However, such experiments would take many months to establish. While they are highly relevant and would undoubtedly strengthen the manuscript, we regard them as directions for future investigation.

Size distribution and quantification of small RNAs expressed in the wildtype and hyrax mutant ovaries were presented to the reviewers.

Other changes to the manuscript:

We reevaluated and modified authors' contributions in the revised manuscript.

Second decision letter

MS ID#: bio.062321R1

MS TITLE: A second genetic screen for gurken mRNA mislocalisation uncovers novel phenotypes of piRNA pathway mutants in *Drosophila*

AUTHORS: Rippei Hayashi; Sophie J. Liddell; Rahnya Taghi; Jessie-Siling Li; Sejal Sathe; Shashank Chary; Azusa Hayashi; S. Mark Wainwright; Sheena Pinchin; David Ish-Horowicz

Article Type: Research Article

I am happy to tell you that your manuscript has been accepted for publication in Biology Open, pending our standard publication integrity checks. It was accepted on 15th October 2025.